# Upregulation of Cytokines and Differentiation of Th17 and Treg by Dendritic Cells: Central Role of Prostaglandin E2 Induced by *Mycobacterium bovis*

**DOI:** 10.3390/microorganisms8020195

**Published:** 2020-01-31

**Authors:** Han Liu, Xuekai Xiong, Wenjun Zhai, Tingting Zhu, Xiaojie Zhu, Yifan Zhu, Yongchong Peng, Yongliang Zhang, Jieru Wang, Huanchun Chen, Yingyu Chen, Aizhen Guo

**Affiliations:** 1The National Key Laboratory of Agricultural Microbiology, Wuhan 430070, China; liuhan18415@163.com (H.L.); xiongxuekai0713@gmail.com (X.X.); wjzhai94@gmail.com (W.Z.); 553238808@webmail.hzau.edu.cn (T.Z.); xiaojie.zhu@Murdoch.edu.au (X.Z.); avander1@163.com (Y.Z.); pengyongchong@webmail.hzau.edu.cn (Y.P.); 13343470442@163.com (Y.Z.); wangjr0317@163.com (J.W.); chenhch@mail.hzau.edu.cn (H.C.); chenyingyu@mail.hzau.edu.cn (Y.C.); 2College of Veterinary Medicine, Huazhong Agricultural University, Wuhan 430070, China; 3Hubei International Scientific and Technological Cooperation Base of Veterinary Epidemiology, Huazhong Agricultural University, Wuhan 430070, China; 4Key Laboratory of Development of Veterinary Diagnostic Products, Key Laboratory of Ruminant Bio-products, Huazhong Agricultural University, Wuhan 430070, China; 5Ministry of Agriculture and Rural Affairs, Huazhong Agricultural University, Wuhan 430070, China

**Keywords:** *Mycobacterium bovis*, PGE2, COX-2, dendritic cells, Th17, Treg, BCG

## Abstract

*Mycobacterium bovis* (*M. bovis*) is a zoonotic pathogen that causes bovine and human tuberculosis. Dendritic cells play a critical role in initiating and regulating immune responses by promoting antigen-specific T-cell activation. Prostaglandin E2 (PGE2)-COX signaling is an important mediator of inflammation and immunity and might be involved in the pathogenesis of *M. bovis* infection. Therefore, this study aimed to reveal the character of PGE2 in the differentiation of naïve CD4^+^ T cells induced by infected dendritic cells (DCs). Murine bone marrow-derived DCs were pre-infected with *M. bovis* and its attenuated strain *M. bovis* bacillus Calmette-Guérin (BCG). Then, the infected DCs were co-cultured with naïve CD4^+^ T cells with or without the cyclooxygenase (COX) inhibitor indomethacin. Quantitative RT-PCR analysis and protein detection showed that PGE2/COX-2 signaling was activated, shown by the upregulation of PGE2 production as well as COX-2 and microsomal PGE2 synthase (mPGES1) transcription in DCs specifically induced by *M. bovis* and BCG infection. The further co-culture of infected DCs with naïve CD4^+^ T cells enhanced the generation of inflammatory cytokines IL-17 and IL-23, while indomethacin suppressed their production. Following this, the differentiation of regulatory T cells (Treg) and Th17 cell subsets was significantly induced by the infected DCs rather than uninfected DCs. Meanwhile, *M. bovis* infection stimulated significantly higher levels of IL-17 and IL-23 and the differentiation of Treg and Th17 cell subsets, while BCG infection led to higher levels of TNF-α and IL-12, but lower proportions of Treg and Th17 cells. In mice, *M. bovis* infection generated more bacterial load and severe abnormalities in spleens and lungs, as well as higher levels of COX-2, mPGE2 expression, Treg and Th17 cell subsets than BCG infection. In conclusion, PGE2/COX-2 signaling was activated in DCs by *M. bovis* infection and regulated differentiation of Treg and Th17 cell subsets through the crosstalk between DCs and naive T cells under the cytokine atmosphere of IL-17 and IL-23, which might contribute to *M. bovis* pathogenesis in mice.

## 1. Introduction

Tuberculosis (TB), which is mainly caused by *Mycobacterium tuberculosis* (*M. tb*) and *Mycobacterium bovis* (*M. bovis*), remains a global zoonotic infectious disease and has killed hundreds of millions of people over the past two centuries [1,2]. The emergence of *M. bovis* as a human pathogen is still not well-understood, but it has been plausibly suggested that the domestication of cattle facilitated close contact with humans, resulting in transmission with the eventual evolution the bovine tuberculosis (bTB) strain of *M. bovis* [3,4]. *M. bovis* is the major causative agent of bTB in a range of animal species, resulting in great global losses to agriculture, whose genome sequence is 99.95% identical to that of *M. tb* [5]. *M. bovis* was also the progenitor of the *M. bovis* bacillus Calmette-Guérin (BCG), which resulted from a deletion of five DNA regions, including 38 Open reading frameworks (ORFs), leading to virulence reduction [6]. Since 1921, BCG has been the only licensed vaccine against human TB, despite it showing variable protection in different populations and regions [7]. Exploring the molecular regulation of immunological events induced by *M. bovis* and BCG would help develop a better understanding of *M. bovis* pathogenesis or BCG protection and is critical for the future development of new diagnostics, therapeutics, and vaccines for tuberculosis.

Dendritic cells (DCs) are professional antigen-presenting cells (APC) that act as a bridge between innate and adaptive immunity, shown by their extraordinary capacity to stimulate the production of subsets Th1, Th2, Th17, and regulatory T cells (Treg) from naïve T cells, which are mainly distinguished by different cytokines, such as IFN-γ, IL-12, TNF-α, IL-4, IL-6, and TGF-β, respectively, or expression patterns of cell surface molecules and transcription factors [8,9,10]. The stimulation of T cells by cross-reactive antigens trigger heterologous immunity. We previously found that *M. bovis* and BCG induced different patterns of cytokine and chemokine production in dendritic cells and differentiation patterns in CD4^+^ T cells [11]. The immune responses of TB is clearly a dynamic one, thus much more knowledge is needed to fully understand the differences that occur in T cell phenotypes and functions.

Prostaglandin E2 (PGE2) is a specific prostaglandin that is synthesized by the collective action of phospholipase A2 and cyclooxygenase (COX) and released from cell membranes. Cyclooxygenase (COX) exists in two isoforms: COX-1 and COX-2. COX-2 is inducible and responsible for the inflammatory effects of prostaglandins [12]. Recent studies in experimental models of tuberculosis have demonstrated that *M. tb* infection induces COX-2 expression and the synthesis of PGE2 in macrophages (Mϕs) [13]. In addition, BCG-induced PGE2 production in DCs serves dual functions: it not only stimulates IL-10 production and limits IFN-γ production, but also enhances the production of IL-23 and IL-17 in T cells to stimulate Th17 differentiation [14]. An upregulated COX-2/PGE2 signaling pathway may cause a dysfunctional immune response that favors the survival and replication of *M.tb*. Therefore, the COX inhibitor indomethacin could initiate an anti-TB chemotherapy effect by significantly downregulating *M.tb*-specific FOXP3^+^ T regulatory cells, reducing cytokine responses and Th1 cell proliferation in patients with active TB [15]. Although there is a determined relationship between PGE2 and *M. tb* or BCG infection in vitro or in vivo (macrophage or human alveolar epithelial cells), the differential production of PGE2 in DCs induced by the infection of virulent *M. bovis* and its attenuated BCG, in addition to its function in mediating specific T cell responses, has not been investigated.

The objective of this study was to analyze the role of the activation of COX2/PGE2 signaling in murine bone marrow-derived DCs infected with *M. bovis* and BCG for stimulating specific T cell responses. We showed that *M. bovis* infection activated the COX2/PGE2 signaling pathway in DCs and promoted the differentiation of naïve CD4^+^ T cells into Th17 and Treg cells by upregulating the secretion of IL-17A and IL-23. This would be significant to understand the pathogenesis of tuberculosis related crosstalk between DCs and naive T cells, bacillus persistence, and develop novel strategies and measures to control tuberculosis in both animals and humans.

## 2. Materials and Methods 

### 2.1. Ethics Statement

This study was carried out in strict accordance with the Guide for the Care and Use of Laboratory Animals, Monitoring Committee of Hubei Province, China, and the protocol was approved by the Committee on the Ethics of Animal Experiments at the College of Veterinary Medicine, Huazhong Agricultural University (permit no. HZAUMO-2016-038). 

### 2.2. Bacterial Culture and DC Preparation

*M. bovis* (ATCC 19210) and BCG Tokyo strain (ATCC 35737) were kindly provided by Dr. Li Chuan-You from the Beijing Tuberculosis and Thoracic Tumor Research Institute, China. Bacterial culture and colony plate counting was operated according to a previous study [11].

The 6- to 8-week-old C57BL/6N female mice were purchased from the Wuhan University Center for Animal Experiment (Wuhan, China). Murine bone marrow-derived DCs were obtained as previously described [11,16].

Briefly, 6- to 8-week-old BALB/c mice were sacrificed, and the femurs and tibias were harvested. Bone marrow was flushed out of the shafts and cells were seeded into Petri dishes with complete RPMI 1640 medium supplemented with 10% heat-inactivated fetal bovine serum (FBS) and recombinant murine granulocyte-macrophage colony stimulating factor (rGM-CSF; 20 ng mL^−1^) (Pepro Tech Inc., Rocky Hill, NJ, USA). The cells were incubated at 37 °C in a humidified incubator with 5% CO_2_ for 10 days with media changes every 3 days. The non-adherent cells were harvested, and DCs were defined by expression of the surface markers (CD11c, CD40, CD80, CD86 and MHC-II) by flow cytometry (FACS Calibur). Fluorescent antibodies (Abs) included either red phycoerythrin (PE)-conjugated antibody (PE-Ab) to CD11c or green fluorescein isothiocyanate (FITC)-labelled antibodies (FITC-Ab) to CD40, CD80, CD86 and MHC-II. Corresponding PE- and FITC-labelled isotypes (eBioscience) were used as controls. The percentage of immature DCs (CD11c^+^, low- to intermediate-level expression of MHC-II, low-level expression of CD80 and CD86, and no expression of CD40) was generally greater than 80%. Non-adherent cells were collected by centrifugation before infection, washed once with incomplete RPMI 1640 medium without FBS and then resuspended in complete RPMI 1640 with 10% FBS without rGM-CSF at a density of 1 × 10^6^ cells mL^−1^.

### 2.3. Detection of Cytokines and CD4^+^ T Cell Differentiation with or without the COX-2 Inhibitor Indomethacin

Infection of DC cells: BCG and *M. bovis* were added at a multiplicity of infection (MOI) of 10 to 24-well plates (Costar, Cambridge, MA, USA) containing DCs with or without 50 mg/mL COX inhibitor Indomethacin (Sigma, St. Louis, MO, USA) and 20 ng/mL exogenous PGE2 (Cascade Biochem LTD, Reading, UK) independently. DC cultures were collected at 6, 12, and 24 h post-infection by centrifugation at 200× *g* for 10 min at 2 to 8 °C. The cultural supernatants were transferred into fresh tubes, immediately followed by filtering through 0.22 μm filters, and were stored at −80 °C for cytokine detection. The cells were washed with PBS three times, and 100 µL cell lysates were added per well for protein extraction or 1 mL TRIzol reagent (Invitrogen, Carlsbad, CA, USA) for RNA extraction, according to the manufacturer’s protocol. 

Infected DCs co-cultured with CD4^+^ T cells: Naïve CD4^+^ T cells were isolated and co-cultured with DCs infected or uninfected with *M. bovis* and BCG according to the procedures as follows. The mouse splenocytes and lymph node cells were collected, and the naïve CD4^+^ T cells were labeled with a biotin-antibody cocktail. These cells were then incubated with anti-biotin magnetic beads (MiltenyiBiotec, Shanghai, China) and separated over LS columns (MiltenyiBiotec, China) following the manufacturer’s procedure to obtain the naïve CD4^+^ T cells. The purity of the sorted T cells should be over 99%.

The naïve CD4^+^ T cells (Th0) were distributed in a 24-well plate at a density of 2 × 10^6^ cells/mL. Then, the DC culture, including the supernatant and cells infected or uninfected with *M. bovis* and BCG, was added to the wells, respectively. In blocking groups, the COX-2 inhibitor indomethacin (Sigma, USA) 4 μL (50 mg/mL) was added before the addition of infected or uninfected DC culture. All groups were further incubated for 24 h. Then, the supernatant and mixed cells were separately collected for the further detection of cytokines and CD4^+^ subsets.

The cytokine levels in the cultural supernatant of each infected group collected as described above were detected, including IL-12, IL-6, IL-23, IFN-γ, IL-4, and IL-17, by ELISA, while the cytokine gene expression in the cells was detected, including IL-1β, IL-6, and TNF-α, by qRT-PCR. Furthermore, CD4^+^ T cells were stained with anti-CD4-FITC and anti-CD25-APC and subjected to intracellular staining with anti-Foxp3-PE (Ebioscience, San Diego, CA, USA) and analyzed by flow cytometry on a FACSCalibur™ cell analyzer (Becton Dickinson, USA). The data were processed with FlowJo Version 7.6 (Becton Dickinson, USA).

### 2.4. Detection of PGE2 with ELISA

After the cells were infected with *M. bovis* and BCG, the concentrations of PGE2 in filtered supernatants of cell culture described above were determined by competitive ELISA (cELISA) using the prostaglandin E2 parameter assay kit (R&D Systems Inc., Minneapolis, MN, USA), according to the product’s instruction. The detection limit of the kit was 16.0–41.4 pg/mL.

### 2.5. Transcription of the Genes Related to PGE2 Biosynthesis and T Cell Differentiation

Total RNA was extracted from cultured cells collected, as previously described, at 6, 12, and 24 h post-infection (PI) using TRIzol reagent (Invitrogen, Carlsbad, USA). The RNA concentration was calibrated between 1.8 and 2.0, using a NanoDrop Spectrophotometer (NanoDrop, Wilmington, DE, USA), to a ratio of optical density at a wavelength of 260 nm (OD_260nm_) to 280 nm (OD_280nm_), and the RNA integrity was evaluated by agarose gel electrophoresis. Then, the RNA was reverse-transcribed by using a reverse transcription kit (Toyobo Co. Ltd., Japan) and the cDNA was subjected to qRT-PCR (quantitative real-time PCR) in an ABI PRISM 7500 Sequence Detector (Applied Biosystems). All pairs of primers designed (Table 1) were commercially synthesized for qRT-PCR to determine the differential gene expression of PGE2 biosynthesis and T cell differentiation after BCG and *M. bovis* infected DCs and CD4^+^ T cells. The β-actin gene was used as an internal reference gene.

The qRT-PCR was performed by using the THUNDERBIRD SYBR qPCR Mix (Toyobo, Japan). The total volume of each reaction was 10 µL, including 50 ng cDNA in 3 µL, 2 µM of each primer in 2 µL, and 5 µL 2× SYBR Green dye. Reactions were programmed in an ABI ViiA^TM^ 7 Real-Time PCR System (Applied Biosystems, USA) as follows: 95 °C for 2 min for the hot-start, followed by 40 cycles of 95 °C for 15 s, 61 °C for 30 s, and 72 °C for 45 s. The fluorescence signal was collected at the end of each elongation step. The internal standard (β-actin gene) and all samples were performed in triplicate. The relative gene transcription was obtained by the following formula: 2^−ΔΔCt^, where Δ*C*_t_ = *C*_t_^target gene^ − *C*_t_^β-actin^, and ΔΔ*C*_t_ = Δ*C*_t_^treatment 1^ − Δ*C*_t_^treatment 2^. A comparison of the transcription levels among different related genes was further performed by taking the lowest transcription of the gene as the reference of 1.

### 2.6. Western Blot Assay of COX and PGES

The expression of β-actin was used as the internal reference. The concentration of total proteins was detected with a bicinchoninic acid (BCA) protein assay kit, according to the product instructions (Thermo Fisher Scientific Inc. San Jose, CA, USA and Pierce Biotechnology, Rockford, IL, USA). The proteins were separated on SDS-PAGE gels and transferred to a PVDF membrane. The membrane was blocked with 5% non-fat dry milk in PBS for 4 h and then incubated overnight with individual probing antibodies respectively, including rabbit monoclonal IgG against β-actin (Santa, USA), COX-1/2, mPGES-1/2, and cPGES (Epitomics, Burlingame, CA, USA). This was followed by incubation with an horse radish peroxidase (HRP)-conjugated secondary antibody for 2 h. Finally, 1–2 mL of ECL detection solution A and B was slowly added and incubated for 2 min at room temperature. The Kodak Image Station was used to detect the chemiluminescence signals. The bands were subjected to the grey level assay with ImageJ.

### 2.7. Mouse Infection and Tissue Bacterial Load

The 6- to 8-week-old C57BL/6N female mice were purchased from the Wuhan University Center for Animal Experiment (Wuhan, China). All animal experiments were conducted in the animal biosafety level 3 laboratory (ABSL-3) at Huazhong Agricultural University. The BCG and *M. bovis* strains cultured to logarithmic phase were injected into mice by the tail vein and consisted of 2 × 10^5^ CFU in 0.2 mL. The mice were euthanized respectively on 1, 4, 7, 14, 21, 28, 35, 49, and 56 days. The lung and spleen tissues were weighted and homogenized in 1 mL PBS (50 mM, pH = 7.4), and each homogenate was 10-fold serially diluted to 10^3^ with PBS. Then, 0.2 mL of each dilution was plated on 7H11 plates and grown for 4–6 weeks until the colonies were suitable for counting by the naked eye.

### 2.8. Tissue Histopathology and Immunohistochemistry Examination

Lung and spleen tissue samples at different time points were fixed in 10% neutralized formalin, embedded in paraffin, and then sectioned. The slices were stained with hematoxylin and eosin (HE) stain and observed by light microscopy. The immunohistochemistry stain was performed to check the expression of IL-17 and Foxp3 in tissues. Briefly, after dehydration, antigen retrieval, and the blocking of intrinsic peroxidase, the sections were first incubated with a mouse polyclonal antibody against Foxp3 (Santa Cruz, CA, USA) or mouse polyclonal antibody against IL-17 (Santa Cruz, USA), and then with the respective HRP (horse radish peroxidase) conjugated goat antibody to mouse IgG (Southernbiotech, Birmingham, AL, USA). After washing, the sections were incubated with 3,3′-diaminobenzidine tetrahydrochloride (DAB)/H_2_O_2_ (Sigma, St. Louis, USA) and the positive cells became a brown color. The sections were photographed with an Olympus photomicroscope (Tokyo, Japan), and data were analyzed by Image-Pro Plus Version 6.0 (Media Cybernetics, Carlsbad, CA, USA).

### 2.9. Detection of the Expression of PGE2-Related Genes and Cytokines and CD4^+^ T Cell Differentiation in Spleens

The spleen of each mouse was divided into three equal parts. The total RNA of one-third of spleen cells in different mice groups scarified as above was extracted to detect the gene expression for PGE2, COX-2, mPGES-1, and cytokines, including IFN-γ, IL-1β, IL-6, and TNF-α, by qRT-PCR. Meanwhile, the transcription factors of Th17 and Treg cells were also detected using qRT-PCR. In addition, another one third of spleen CD4^+^ T cells were stained with an anti-mouse CD4-FITC monoclonal antibody and intracellularly stained with an anti-mouse-IL-17A-PE monoclonal antibody (Ebioscience, Waltham, MA, USA). The final one-third of spleen CD4^+^ T cells were stained with an anti-mouse-CD4-FITC and anti-mouse-CD25-APC monoclonal antibody and subjected to intracellular staining with an anti-mouse-Foxp3-PE monoclonal antibody (eBioscience, Waltham, MA, USA). All the stained CD4^+^ T cells were analyzed by flow cytometry on a FACSCalibur™ cell analyzer (Becton Dickinson, San Jose, CA, USA), and data were processed with FlowJo Version 7.6 (Becton Dickinson, USA).

### 2.10. Statistical Analysis

The data were expressed as means ± the standard deviation (SD) of the mean. The two-tailed unpaired *t*-test was used for a comparison of two groups, one-way analysis of variance (ANOVA) for comparisons of three or more groups, and Pearson’s chi-squared test for categorical variables. Values of *p* < 0.05 were considered to be significantly different (*) and *p* < 0.01 very significantly different (**).

## 3. Results

### 3.1. M. bovis and BCG Infection Differentially Upregulated the COX2/PEG2 Pathway in DCs

In the present study, to clearly understand the role of Prostaglandin (PG) E2 in *mycobacteria* infected murine dendritic cells (DCs), the virulent *M. bovis* and the attenuated BCG strains were used in experiments. More than 80% immature DCs were defined by expression of the surface markers (CD11c, CD40, CD80, CD86 and MHC-II) by flow cytometry (Figure 1). 

To demonstrate the role of PGE2 in regulating responses of DCs to *M. bovis*/BCG infection, PGE2 concentrations were quantified in the supernatants of DC cultures with or without *M. bovis*/BCG infection at 6, 12, and 24 h post-infection (PI). It was demonstrated that PGE2 production was enhanced in both infected DC groups compared to that in the uninfected controls; more importantly, PGE2 concentrations were significantly higher in *M. bovis*-infected groups than in the BCG-infected groups (*p* < 0.01). Furthermore, the PGE2 concentrations from *M. bovis*-infected groups were increased at 6 h, 12 h and then 24 h PI in *M. bovis*-infected groups, but changed little in the BCG-infected group at the three time points (Figure 2A), indicating that virulent *M. bovis* is vital for the accumulation of PGE2 in the bacteria-infected DCs in vitro. The mRNA and protein levels of COX-1, COX-2, microsomal PGE2 synthase 1/2 (mPGES-1/2) and cPGES involved in PGE2 biosynthesis were measured in *M. bovis-* and BCG-infected DCs. Although the expression of COX-1 (Figure 2B), mPGES-2 (Figure 2E), and cPGES (Figure 2F) exhibited no significant difference between the two infected groups and was even lower than the control, the mRNA levels of COX-2 (Figure 2C) and mPGES-1 (Figure 2D) were significantly increased in DCs infected by both mycobacterial species compared with the uninfected control (*p* < 0.01). Correspondingly, the protein levels of COX-2 and mPGES-1 were also significantly enhanced, whereas COX-1, mPGES-2, and cPGES only slightly changed (Figure 2G,H). These results suggest that both *M. bovis* and BCG infection were able to enhance PGE2 production, likely through upregulating the expression of COX-2 and mPGES-1 in DCs, though a stronger effect was observed for *M. bovis*. 

To further determine whether the COX2/PGE2 pathway was specifically activated, we compared *M. bovis*- and BCG-infected DCs with and without the COX inhibitor indomethacin. Indomethacin not only markedly reduced the expression of COX-2 (Figure 3A, *p* < 0.01) and mPGES-1 (Figure 3B, *p* < 0.01) compared to that in the infected groups without indomethacin, but also removed the difference in COX-2 and mPGES-1 expression between *M. bovis*- and BCG-infected DCs.

On the other hand, the actions of PGE2 are mediated by a series of E-prostanoid (EP) receptors, which can be divided into four pharmacological classes, classified as EP1 to EP4 [17]. The quantitative analysis conducted by real-time RT-PCR showed that *M. bovis-*/BCG-infected DCs expressed high levels of EP2 (Figure 3D) and EP4 (Figure 3F) mRNA, whereas EP1 (Figure 3C) and EP3 (Figure 3E) mRNA expression was low or below the detection limit. Furthermore, treatment with indomethacin attenuated this observed upregulated expression of EP2 and EP4 in DCs induced by *M. bovis*/BCG infection, which were reduced to similar levels as uninfected and untreated controls. These results indicated that the expression of EP2 and EP4 receptors was mediated by the activation of the COX-2/PGE2 pathway in DCs infected by *M. bovis* and BCG. 

### 3.2. COX-2 Inhibitor Suppressed the Differentiation of Naïve CD4^+^ T Cells Induced by M. bovis- and BCG-Infected DCs

PGE2 is an important inflammatory mediator that is derived from arachidonic acid through the cyclooxygenase (COX1 and COX2) biosynthesis pathway [18]. In order to investigate the relationship between PGE2 and cytokine production, IL-1β, IL-6, and TNF-α were measured by qRT-PCR, whereas IL-6, TNF-α,IL-12 and IL-23 were measured by ELISA, and these were obtained from DCs cultures with or without the infection of either *M. bovis* or BCG after PGE2 was inhibited by the COX inhibitor. The protein expression of IL-6 (Figure 4B,D) and TNF-α (Figure 4C,E) obtained from the ELISA approach was in accordance with these ratios from qRT-PCR analysis. In both *M. bovis*- and BCG-infected groups, TNF-α, IL-12, and IL-23 were significantly reduced after PGE2 blocking (*p* < 0.01, Figure 4E,F,G). On the contrary, IL-1β was increased after PGE2 blocking in infected groups (*p* < 0.01, Figure 4A). 

To test whether PGE2 could promote differentiation of DCs, we co-incubated DCs with COX inhibitor-indomethacin or PGE2. Expression of DC surface Ag CD11c (Figure 5A,B) and major histocompatibility complex class II (MHC II) (Figure 5C,D) in different DCs groups was evaluated using flow cytometry. Compared with CD11c, we found that exogenous added PGE2 significantly increased the number of MHC Ⅱat 24 h infection. Our data indicate that PGE2 can increase expression of MHC Ⅱ to activated DCs, suggesting the potential role of these antigen-presenting cells in the bioactive lipid-induced immune response.

Numerous studies have indicated that control of *Mycobacterium tuberculosis* (*M.tb*) infection requires CD4+ T-cell responses and MHC II presentation of *M.tb* antigens (Ags) [11,19]. The mouse splenocytes and lymph node cells were collected and separated to obtain the naïve CD4^+^ T cells. Then, these cells were respectively co-cultured for 24 h with the infected DCs culture, with or without the COX inhibitor indomethacin. The naïve CD4^+^ T cells were co-cultured with mycobacteria-infected DCs, and the transcriptional expression of T-bet, GATA3, Foxp3, and ROR-γt, which are critical markers for Th1, Th2, Treg, and Th17 cells, respectively, was measured using qRT-PCR. The results showed a significant increase in the mRNA levels of GATA3, Foxp3, and ROR-γt in co-cultured CD4^+^ T cells after *M. bovis* infection, and an even greater decrease in the level of T-bet (Figure 4H). 

Furthermore, indomethacin treatment reduced the expression of IFN-γ and IL-17A (Figure 4I,K) from differential high levels to similarly low levels between the two bacterial groups of DCs co-cultured with CD4^+^ T cells. In other words, the differential expression of IFN-γ and IL-17A in *M. bovis*-/BCG-infected DCs co-cultured with naïve CD_4_^+^ T cells was mostly regulated by the COX2/PGE2 signaling pathway with a pattern of upregulating IL-12, IL-23, and TNF-α expression, while downregulating IL-1β expression. Taken together, these results indicated that *M. bovis* might trigger the expression and secretion of pro-inflammatory cytokines, IL-17A and IL-23, resulting in PGE2 accumulation increasing inflammatory damage and adjusting the immune response.

Then, CD4^+^ T cell subsets were gated and the proportion of Treg (CD25^+^Foxp3^+^) induced by mycobacteria-infected DCs was analyzed with flow cytometry. The stimulation of *M. bovis*-infected DCs yielded a significant increase in Treg, with the proportion of 6.92% (*p* < 0.01). BCG-infected DCs displayed a significantly weaker ability to induce Treg differentiation than *M. bovis* (*p* = 2.3E-09), and the proportion was 4.26%. By contrast, both proportions of BCG- and *M. bovis*-infected DCs showed a significant decrease in the ability to stimulate the development of Treg after PGE2 blocking (*p* < 0.01) (Figure 4L). Additionally, there were no significant differences in the Treg differentiation of naïve CD4^+^ T cells between the *M. bovis*-infected and BCG-infected groups treated with the COX2 inhibitor (Figure 4L). Based on the effect of the COX-2 inhibitor indomethacin on the expression of cytokines and transcription factors by CD4^+^ T cells induced by mycobacteria-infected DCs, it was concluded that COX-2 signaling significantly regulated the differentiation of Treg (represented by FoxP3 expression), Th1 (represented by the expression of T-bet, IL-12, TNF-α, and IFN-γ), and Th17 (represented by the expression of IL-17 and IL-23). All tested cytokines were significantly elevated, which was consistent with the results of T-bet transcription (Figure 4H). The development of these CD4^+^ T cell subsets was mediated by COX-2/PEG2 signaling upon DC infection. Collectively, the data presented in Figure 4 indicate that *M. bovis*-infected DCs tend to be involved in the differentiation of Th2, Treg, and Th17 cells, but BCG-infected DCs were inclined to induce Th1 cell differentiation. 

### 3.3. Higher Bacterial Load and More Severe Tissue Lesions Developed by M. bovis Infection Than BCG in Mice 

The mycobacterial burden and histopathological lesions in the lung and spleen tissues of infected mice were used to comparatively assess the virulence of *M. bovis* and BCG in mice. The strains cultured to logarithmic phase were injected into mice by the tail vein, consisting of 2 × 10^5^ CFU per mouse, and mice were then respectively sacrificed on 1, 4, 7, 14, 21, 28, 35, 49, and 56 days post-infection. In the spleens from *M. bovis* and BCG infection groups, the bacterial load increased slowly after the first day, maintained a high level for 14 days, and then began to decline (Figure 6A). The bacterial load in the lungs of the *M. bovis* infection group exhibited a similar trend after the first day (*p* < 0.01) but maintained the high level for a longer time and decreased from 21 days, then remained relatively stable after 42 days (Figure 6B). Meanwhile, the amount of *M. bovis* was significantly higher than BCG in both the lungs and spleens (*p* < 0.01). In addition, the spleen bacterium load was significantly higher than the lung bacterium load.

Visual inspection showed that only spleens were enlarged, while the lungs, livers, and kidneys displayed no apparent macroscopic lesions (data not shown). Next, the damage in these tissues was assessed by H&E staining. As shown in Figure 5, both *M. bovis*- and BCG-infected groups induced a histopathological response in the mouse spleen, shown by the increased macrophages, reduced lymphocytes, and congestion of the red pulp zone (Figure 6C). The abnormality caused by *M. bovis* in the lungs included a thicker pulmonary alveolar wall and infiltration of the alveolar cavity by macrophages and lymphocytes. In the BCG infection group, although a similar histopathology occurred, the severity was less than in the *M. bovis* group (Figure 6D). Therefore, these histopathological observations were consistent with the results of bacterial burdens in the lungs and spleens from infected mice.

### 3.4. Expression of COX-2, mPGES1, and Cytokines was Elevated in Spleens of Infected Mice 

Because the spleens of mice infected with *M. bovis* had a higher bacterial load and more severe histopathological change than those of BCG animals, we next investigated whether COX-2-PGE2 signaling and the expression of the above cytokines were altered in spleens.

The total RNA of spleens from different groups of mice was extracted and used to detect PGE2-associated signaling molecules and cytokines, including COX-2 (Figure 7A), mPGES-1 (Figure 7B), IFN-γ (Figure 7C), IL-1β (Figure 7D), IL-6 (Figure 7E), and TNF-α (Figure 7F), by qRT-PCR. During the infection, the expression of COX-2 in the *M. bovis*-infected group was significantly higher than that in the BCG-infected group at all time points, with arrival at the peak occurring at day 14 and 21 PI (*p* < 0.01) (Figure 7A). Although the difference in mPGES-1 expression was significant between the two infection groups at all time points, only at some time points (day 14, 21, 28, 49, and 56 PI) was the expression of mPGES-1 in the *M. bovis*-infected group significantly higher than that in the BCG-infected group (*p* < 0.01). Similar to COX-2, mPGES-1 expression reached the peak at day 14 PI (Figure 7B). In addition, both infected groups displayed a significantly higher expression of COX-2 and mPGES-1 than the uninfected control. 

Generally speaking, the cytokines increased gradually with time, reached the peaks during the period of day 7 and 21 PI, and then decreased, but the differences always existed between *M. bovis-* and BCG-infected groups for each cytokine. Besides, IL-1β production displayed another early peak at day 1 PI. Similar to the findings from cell culture, *M. bovis*-infected mice exhibited higher levels of IL-1β (during day 1 to 14 PI) and IL-6 (during day 1, 7, 14, and 49 PI), but lower levels of TNF-α (Figure 7F) (during day 1 to 21 PI) than those in BCG-infected mice (*p* < 0.05), but at other time points the opposite tendency occurred. Contrary to the findings from the cell culture, the expression of IFN-γ in the *M. bovis*-infected group was significantly higher than that in the BCG-infected group at most sampling time points (*p* < 0.01) (Figure 7C). 

Serum concentrations of the cytokines IFN-γ, IL-6, IL-1β, and TNF-α were further detected (Figure 8). As a result, all cytokines could be significantly elevated in the infected mice at some time points compared with the control mice and there was a significant difference between *M. bovis*- and BCG-infected groups at some time points. However, there were no consistently higher or lower groups in any cytokines during the observation period from day 1 to 56 PI. Surprisingly, when a cross-sectional comparison of these four cytokines was performed, it was found that the levels of IL-6, IL-1β, and TNF-α at day 28 PI and IFN-γ at day 21 PI in the BCG-infected group were significantly higher than in the *M. bovis*-infected group (*p* < 0.01), while at day 49 PI, the levels of all four cytokines in the BCG-infected group were significantly lower than the *M. bovis*-infected group (*p* < 0.01), indicating that the production of cytokines stimulated by *M. bovis*/BCG infection occurred in a time-dependent way. In addition, there is only partial agreement in the difference of cytokine production between *M. bovis*- and BCG-infected groups among assays of the cell culture (Figure 4), the qRT-PCR of spleens (Figure 7), and the concentration of serum cytokines (Figure 8). In the *M. bovis*-infected group, the production of IL-1β expression on day 1 PI and IL-6 expression on day 14 and 49 PI was significantly higher than in the BCG-infected group (*p* < 0.01) in terms of both levels of serum protein and spleen transcription, in agreement with the findings from the cell culture. In addition, the transcription levels of TNF-α on day 1–21 PI and serum concentrations of TNF-α on day 28 PI were significantly higher in the BCG-infected group than the *M. bovis*-infected group, similar to the findings from the cell culture (*p* < 0.01). Although there are some time points when the IFN-γ expression was higher in the BCG-infected group than the *M. bovis*-infected group, the results were similar to the findings from the cell culture (*p* < 0.01) in that the *M. bovis*-infected group produced higher levels of IFN-γ at most time points.

### 3.5. Differentiation of Treg and Th17 Cells in Mouse Spleens after Mycobacterial Infection 

To study whether the differentiation of T cells was affected by *M. bovis* and BCG infection in mice, the proportion of Treg (CD25^+^Foxp3^+^) and Th17 (CD4^+^ IL-17A^+^) and their specific transcription factors Foxp3 or ROR-γt in spleens were detected.

As shown in Figure 9A, the expression of Foxp3 and differentiation of Treg in both *M. bovis*- and BCG-infected groups increased significantly (*p* < 0.01) and reached peak values on day 1 (Figure 9A-a), while *M. bovis* infection stimulated a higher expression than BCG infection (*p* < 0.01). Then, Foxp3 transcription levels in both infected groups decreased to some extent but were maintained during the period of 4 to 14 days. From day 21 to 56, Foxp3 transcription decreased to the basic level of the negative control, but the *M. bovis*-infected group was significantly higher than the BCG-infected group on days 21, 28, 42, and 56 (*p* < 0.01). Furthermore, the difference trend in proportions of Treg cells between *M. bovis*- and BCG-infected groups was similar to Foxp3 transcription but was maintained for longer than Foxp3 transcription (from day 1 to 7) (*p* < 0.01) (Figure 9A-b). To assess the effects of *M. bovis* and BCG infection on Foxp3 expression in spleens, we performed the immunohistochemical staining of Foxp3 expression in spleen tissues sampled at day 4 PI. Random fields in each picture were quantitatively evaluated by Image-Pro Plus and the results revealed that *M. bovis* infection significantly increased the percentage of Foxp3^+^ cells in spleens (Figure 9A-c,d).

Additionally, both *M. bovis* and BCG infection obviously upregulated the transcriptional levels of ROR-γt from day 1 to 28 PI (Figure 9B-a) and IL-17 from day 1 to 42 days (Figure 9B-b), and reached the peak value on day 14PI for ROR-γt and day 7PI for IL-17 (*p* < 0.01). Then, they began to decrease, and *M. bovis* infection induced a higher expression of ROR-γt and IL-17 representing a Th17 subset. Further assessment of Th17 (CD4^+^ IL-17A^+^) proliferation (Figure 9B-c) in the spleen revealed that both *M. bovis* and BCG infection stimulated the highest proliferation of Th17 cells in infected mice at day 14 PI. Then, the proportion of Th17 was gradually increased, but *M. bovis* infection induced a higher proportion of Th17 cells at a later stage thereafter (at day 21, 28, and 49 PI). Taken together, the in vivo infection evidence was consistent with that from in vitro cellular experiments, which indicates that *M. bovis* induced a significantly greater differentiation of Treg and Th17 cells than BCG. 

## 4. Discussion

We previously reported that *M. bovis* infection induces different patterns of cytokine in dendritic cells and differentiation patterns in CD4^+^ T cells [11]. In the present study, we investigated the role of PGE2 in DCs infected with a virulent *M. bovis* or the derived attenuated BCG. Our data clearly demonstrated that PGE2 induced by *M. bovis* promotes the differentiation of naïve CD4+ T cells into Th17 and Treg upon DCs infection, as based on the following lines of evidence: (i) DCs upregulates PGE2 production in response to *M. bovis* infection; (ii) the COX-i indomethacin exhibits reduced production of IL-17A and IL-23 markedly triggered by *M. bovis*; (iii) the exogenous added PGE2 significantly increased the number of MHC Ⅱ in DCs, the COX-i indomethacin exerted an obvious decrease in Treg which was highly stimulated by *M. bovis* infected DCs; (iv) C57BL/6N mice reveal significant elevation of expression of COX2 and mPGES1 and also the transcription of cytokines after *M. bovis* infection; and (v) *M. bovis* infection leads to a higher level of differentiation of Treg and Th17 cells in murine spleen, resulting in enhanced bacterium load and lesions.

### 4.1. M. bovis and BCG Infection Specifically Activated COX-2/PGE2 Signaling in DCs

In the current study, *M.bovis* induced higher PGE2 production since 6 h PI than BCG in DCs. COX-1 is a typical constitutive gene that is lack of TATA box, and COX-2 is an inducible enzyme that becomes abundant in activated macrophages and other cells at sites of inflammation. Previous study confirmed that BM-DC constitutively expressed COX-1 but not COX-2, while PGE2 derived largely from COX-2 by mycobacteria because of the LPS in a dose-dependently fashion [20,21]. The present study has uncovered that both BCG and *M.bovis* induced higher COX-2 but not COX-1, indicating that COX-2 was the predominant isoform of COX involved in PGE2 production after infected by mycobacteria. Furthermore, mPGES-2 and cPGES are constitutively expressed genes and mPGES-1, a membrane-associated perinuclear protein, seems to be preferentially functionally coupled with COX-2 [22]. In the current study, both BCG and *M.bovis* induced higher mPGES-1 mRNA transcription levels, and this trend was exactly the same as COX-2 expression, indicating that the strong production of PGE2 in DCs after infection is mediated by both COX-2 and mPGES-1. The activation of COX-2/PGE2 signaling-regulated chronic infections and cancer has been supported by several lines of evidence, including the following: influence of the apoptosis of colon cancer (CC) cells and the prognosis of CC patients via establishing a PVT1/miR-146a/COX2 signaling pathway [23,24]; attenuation of chronic antiviral T-cell responses through constitutive COX2-dependent PGE2 synthesis by lymph node fibroblasts [25]; regulation of PD-L1 expression in tumor-infiltrating myeloid cells through the COX2/mPGES1/PGE2 pathway [26]; inhibition of COX2, which profoundly affects the suppressive function of the established MDSCs isolated from ovarian cancer patients [27]; and disruption of early stages of DC differentiation and reinforcing the migratory function of DCs through mPGES-1 [28].

Prostaglandins are important mediators of inflammation [29]. PGE2 is synthesized from COX-2-derived PGH2 and functions through four G-protein-coupled E-type prostanoid (EP) receptors, namely, EP1, EP2, EP3, and EP4, each of which can activate different downstream signaling pathways [30]. In addition, it was demonstrated that PGE2 makes a contribution to mycobacterial infection by *Mycobacterium leprae* [31] and *M. tb* [14,32]. Similar results were obtained in this study. For example, we found that *M. bovis* and BCG infection induced the production of cytokines, including IL-1β, IL-6, TNF-α, IL-12, IFN-γ, IL-4, IL-17A, and IL-23, but the COX inhibitor indomethacin completely suppressed these effects in DCs. Therefore, it was demonstrated that COX-2 signaling regulated the production of TNF-α, IL-12, IL-17, and IL-23. In addition, BCG infection induced higher levels of TNF-α and IL-12, while *M. bovis* infection induced higher levels of IL-17 A and IL-23. When the infected DCs were co-cultured with CD4^+^ naïve T cells, the *M. bovis* infection was inclined to induce the differentiation of Th17 and Treg represented by the corresponding cytokine markers or transcription markers. Taken together, these results strongly suggest that the inhibition of PGE2 signaling would be a very important tool for controlling the pathogenesis of mycobacterial induced chronic inflammatory diseases.

### 4.2. Differences in Differentiation of Treg and Pathogenic Th17 Contribute to Pathogenesis Stimulated by Mycobacterium bovis

Infection with *M. bovis* and BCG stimulated the differentiation of naïve CD4^+^ T cells into subsets of Th17 and Treg in the mouse model with the corresponding cytokine atmosphere, and COX-2/PGE2 signaling mediated these processes. As in the cell culture, cytokine expression was consistently induced during the interaction of infected DCs and naïve CD4^+^ T cells at most time points (Figure 10). However, there was no consistent change tendency at all time points for both *M. bovis*- and BCG-infected groups. This could result from the animal and *in vivo* bacterial status [33] during the 56 days of the observation period.

Previously, it has been well-established that the overexpression of COX-2 and PGE2 contributes to pathogenesis in a number of bacterial, fungal, and viral infections, such as *Streptococcus suis* [34], *Escherichia coli* [35,36], *Chlamydia trachomatis* [37], *Candida*
*albicans* [38], H1N1 influenza virus [39], and Theiler’s murine encephalomyelitis virus [40]. Correspondingly, the bacterial load and abnormality levels in *M. bovis*-infected mice were higher than those in BCG-infected mice. Therefore, it was concluded that COX-2/PGE2 signaling upregulated the development of Treg and Th17 cell subsets induced by DCs infected with *M. bovis*, which likely contributed to the higher levels of bacterial load and abnormalities in mouse spleens and lungs. It is reasonable to speculate that the excessive concentrations of PGE2 induced by *M.bovis* infection mediate stronger tissue destructive forms of immunity. However, IFN-γ production did not show an apparent association with COX-2/PGE2 signaling, and inconsistent results were obtained for the cell model and mouse model. The reason for this remains to be investigated in the future.

In conclusion, COX-2/PGE2 signaling was differentially activated by *M. bovis* and BCG infection in infected DCs and CD4+ T cells and further regulated the differentiation of naïve CD4+ T cells into Treg and Th17 cell subsets induced by infected DCs, which might contribute to the over inflammation induced by *M. bovis* infection compared to BCG.

## Figures and Tables

**Figure 1 microorganisms-08-00195-f001:**
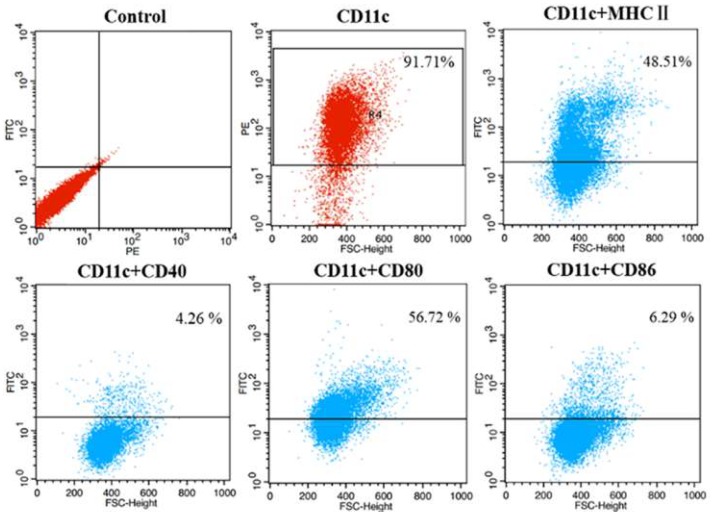
Analysis of cellular surface Ag expression on murine bone marrow derived dendritic cells. The immature dendritic cells (DCs) were defined by expression of the specific surface markers (CD11c, CD40, CD80, CD86 and MHC-II) by flow cytometry.

**Figure 2 microorganisms-08-00195-f002:**
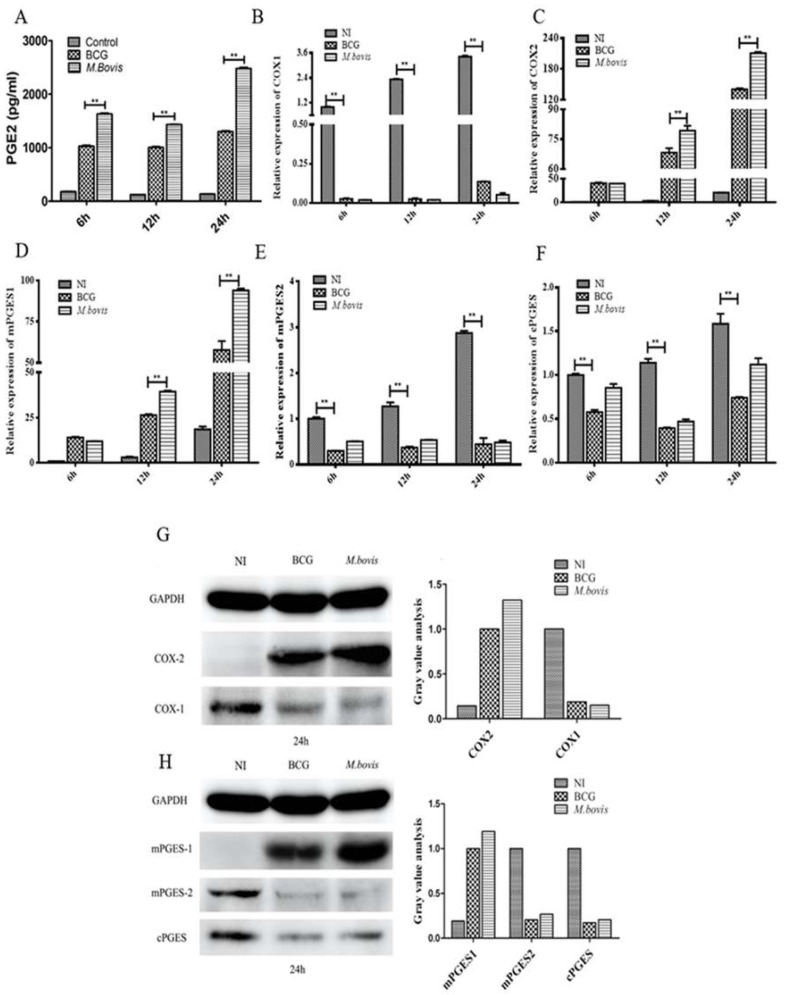
*Mycobacterium bovis* (*M. bovis*) more strongly activated cyclooxygenase (COX)2-prostaglandin E2 (PGE2) pathways in dendritic cells (DCs) than compared to *M. bovis* bacillus Calmette-Guérin (BCG). DCs were infected with BCG and *M. bovis* at a multiplicity of infection (MOI) of 10. (**A**) PGE2 concentrations were measured with commercial ELISA kits at 6, 12, and 24 h post-infection (PI). The relative transcription of COX-1 (**B**), COX-2 (**C**), microsomal PGE2 synthase 1 (mPGES-1) (**D**), microsomal PGE2 synthase 2 (mPGES-2) (**E**), and cPGES (**F**) was measured by real-time RT-PCR at 6, 12, and 24 h PI. The transcription activity of infected DCs was normalized against the uninfected DC control, whose β-actin transcription activity was taken as 1. Protein expression of COX-1/2 (**G**), mPGES-1/2, and cPGES (**H**) was measured with Western blotting and gray value analysis at 24 h PI. The data shown are the mean ± SD of three replicates, where * represents *p* < 0.05 and ** represents *p* < 0.01.

**Figure 3 microorganisms-08-00195-f003:**
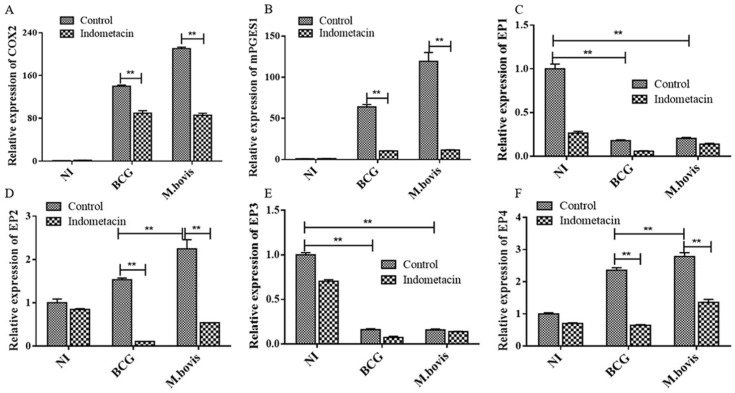
The effect of the COX inhibitor indomethacin on the expression of COX-2, mPGES1, and PGE2 receptors. DCs were infected with *M. bovis* and BCG at an MOI of 10. The mRNA expression of COX2 (**A**), mPGES1 (**B**), and E-prostanoid (EP)1-4 (**C**–**F**) in different DC groups with or without 50 mg/mL indomethacin treatment. The data shown are the mean ± SD of three replicates, where * represents *p* < 0.05 and ** represents *p* < 0.01.

**Figure 4 microorganisms-08-00195-f004:**
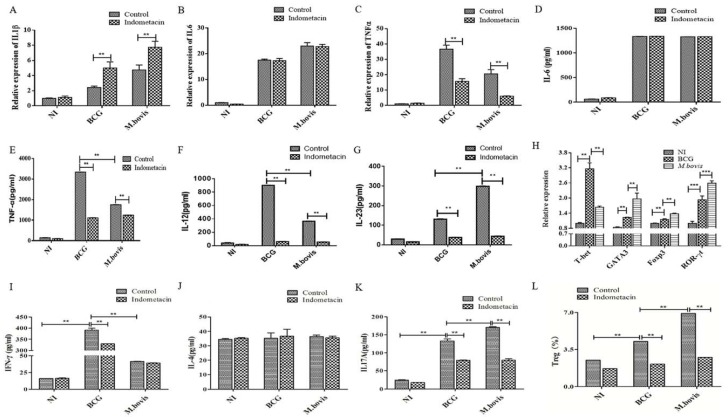
The effect of the COX inhibitor indomethacin on the differentiation of CD4^+^ T cell subsets stimulated by mycobacteria-infected DCs. DCs were infected with BCG and *M. bovis* for 24 h at a multiplicity of infection (MOI) of 10, and 50 mg/mL indomethacin was added independently. The relative transcription of IL-1β (**A**), IL-6 (**B**), and TNF-α (**C**) in DCs was measured with real-time RT-PCR. The concentrations of IL-6 (**D**), TNF-α (**E**), IL-12 (**F**) and IL-23 (**G**) in DCs were measured with commercial ELISA kits at 24 h post-infection (PI). The DCs infected with *M. bovis* and BCG for 24 h at an MOI of 10 were co-cultured with naïve CD4^+^ T cells for 24 h. The DCs without mycobacteria infection/naïve CD4^+^ T cells (**NI**) were employed as a negative control. Infected group: CD4^+^ T cells co-cultured with an *M. bovis*- or BCG-infected DC culture. Inhibition group: COX inhibitor indomethacin was added to the culture. (**H**) Transcriptional levels of T-bet, GATA3, Foxp3, and ROR-γt were measured with qRT-PCR. The secretion proteins of IFN-γ (**I**), IL-4 (**J**), and IL-17A (**K**) were measured with ELISA kits. (**L**) Proportion of Treg (CD25^+^Foxp3^+^) was measured with flow cytometry. The data are shown as the mean ± SD of three replicates, where * represents *p* < 0.05 and ** represents *p* < 0.01.

**Figure 5 microorganisms-08-00195-f005:**
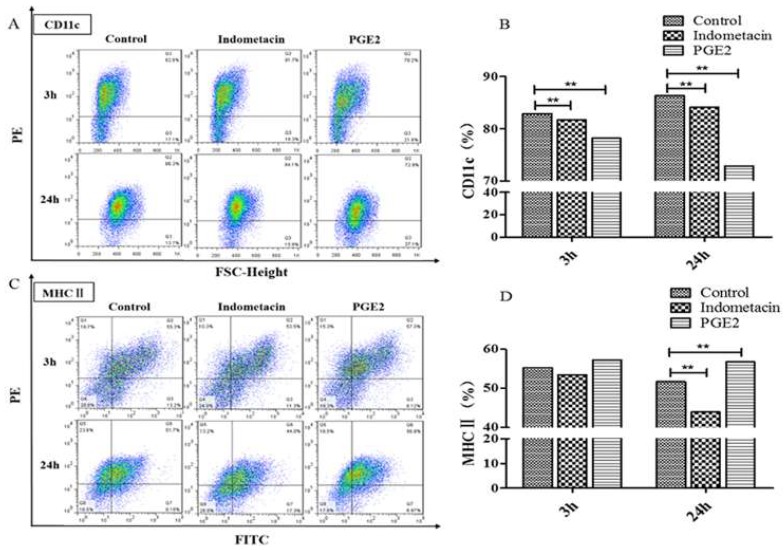
The differentiation of DCs associated with PGE2. DCs were co-incubated with 50mg/mL indomethacin and 20ng/mL exogenous PGE2 independently. Expression of DCs surface Ag PE- CD11c (**A** and **B**) and FITC-MHC II (**C** and **D**) were measured with flow cytometry.

**Figure 6 microorganisms-08-00195-f006:**
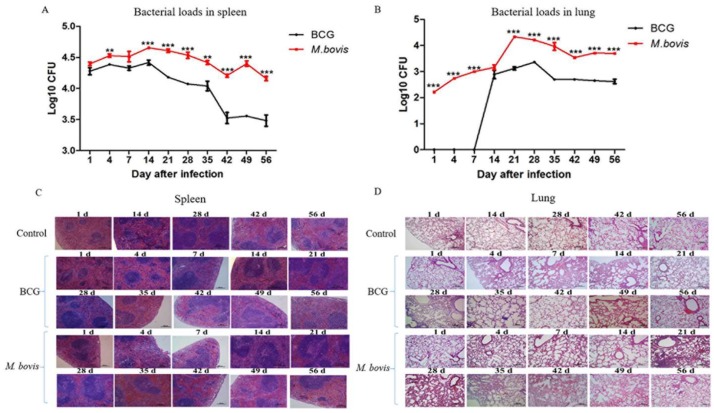
Bacterial loads and histopathological change in the spleens and lungs of mice infected with *M. bovis* and BCG. Mice were infected with *M. bovis* and BCG by tail vein injection with 2 × 10^5^ CFU/mouse and sacrificed on 1, 4, 7, 14, 21, 28, 35, 49, and 56 days. (**A**) The bacterial load (CFU) in the spleens of infected C57BL/6N mice. (**B**) The bacterial load (CFU) in the lungs of infected C57BL/6N mice. The zero (0) on the *y*-axis indicates that no colonies were obtained from infected mice. (**C**) Histopathological change in the spleens (H&E staining, 100×). (**D**) Histopathological change in the lungs (H&E staining, 100×). The data are shown as the mean ± SD of three replicates, where *, **, and *** represent *p* < 0.05, *p* < 0.01, and *p* < 0.001, respectively.

**Figure 7 microorganisms-08-00195-f007:**
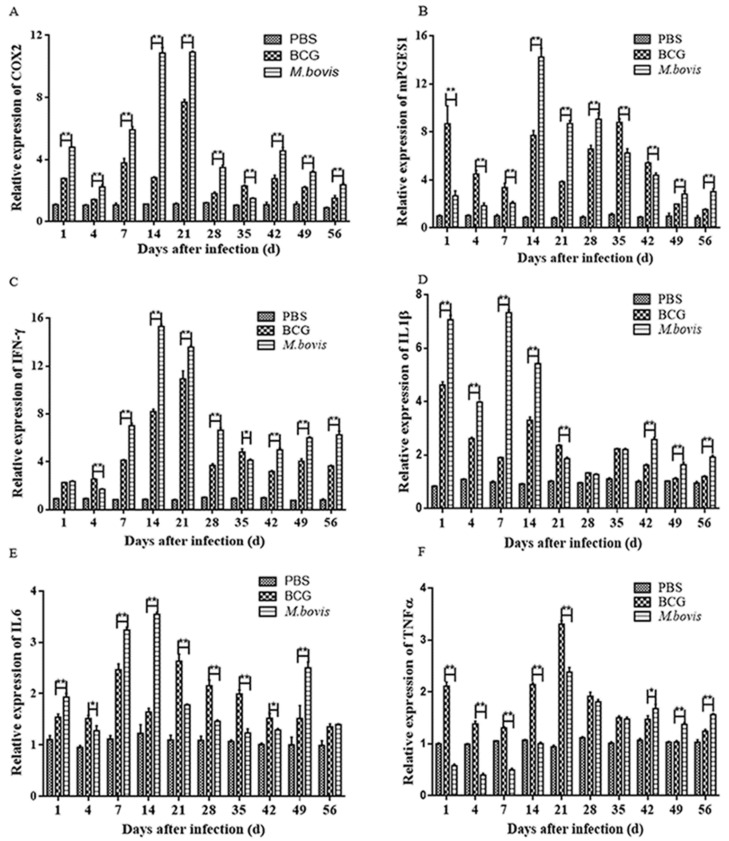
The differential mRNA transcription levels of PGE2 biosynthesis-related genes and inflammatory cytokines in the spleens of mice infected by *M. bovis* and BCG. The mice were infected with *M. bovis* and BCG by tail vein injection with 2 × 10^5^ CFU/mouse and sacrificed on 1, 4, 7, 14, 21, 28, 35, 49, and 56 days post-infection. Real-time PCR was used to determine the mRNA expression level for COX-2 (**A**), mPGES-1 (**B**), IFN-γ (**C**), IL-1β (**D**), IL-6 (**E**), and TNF-α (**F**) in the spleens of C57BL/6N mice. The relative expression levels were expressed as 2^-ΔCT.^ The data are shown as the mean ± SD of three replicates, where * represents *p* < 0.05, ** represents *p* < 0.01, and *** represents *p* < 0.001.

**Figure 8 microorganisms-08-00195-f008:**
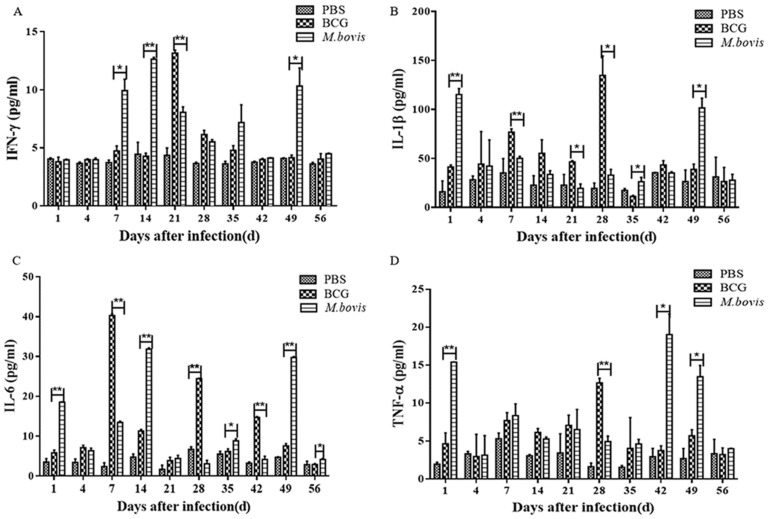
Serum cytokine production in mice infected with *M. bovis* and BCG. The mice were infected by tail vein injection with 2 × 10^5^ CFU each and euthanized on 1, 4, 7, 14, 21, 28, 35, 49, and 56 days post-infection. The serum concentrations for IFN-γ (**A**), IL-1β (**B**), IL-6 (**C**), and TNF-α (**D**) were determined with commercial ELISA. The data are shown as the mean ± SD of three replicates, where *, **, and *** represent *p* < 0.05, *p* < 0.01, and *p* < 0.001, respectively.

**Figure 9 microorganisms-08-00195-f009:**
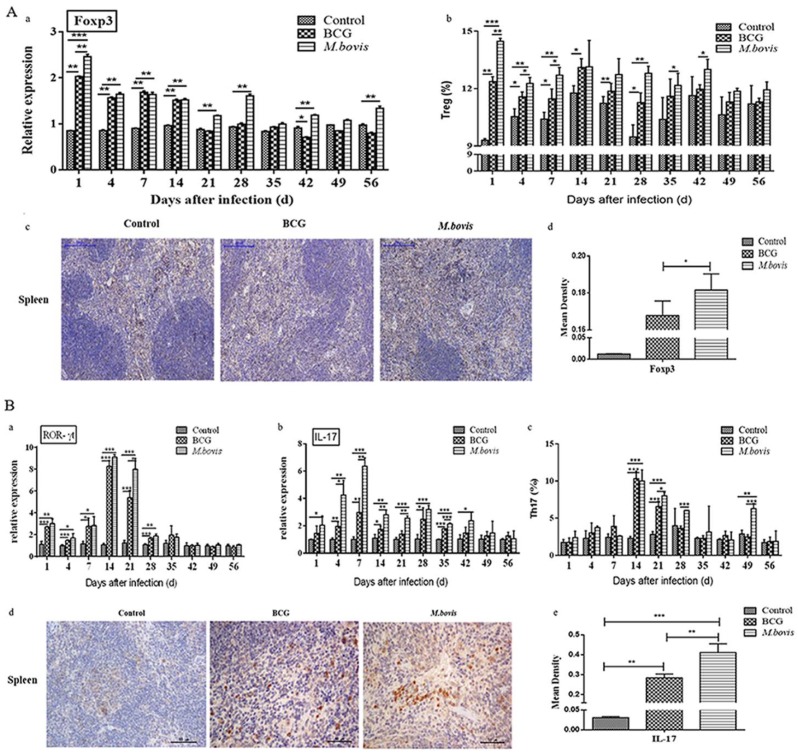
Differentiation of CD4^+^ Treg and Th17 cell subsets in mice induced by *M. bovis* and BCG infection. Treg (**A**) and Th17 (**B**) cells in the spleen of *M. bovis*- and BCG-infected mice. Mice were infected with *M. bovis* and BCG by tail vein injection with 2 × 10^5^ CFU each and sacrificed on 1, 4, 7, 14, 21, 28, 35, 49, and 56 days PI. The total RNA of spleens in different mice groups was collected to detect transcription factors, including Foxp3 (**A-a**), ROR-γt (**B-a**), and IL17 (**B-b**), by qRT-PCR. Splenocytes were gated, and the proportion of Treg (CD25^+^Foxp3^+^) (**A-b**) as well as Th17 (CD4^+^ IL-17A^+^) (**B-c**) was analyzed with flow cytometry and then calculated by analyzing FACS data. The expression of Foxp3 (**A-c**, 100×) and IL17 (**B-d**, 400×) was measured with immunohistochemistry (IHC). Blue indicates cellular nuclei and yellow represents a positive signal of Foxp3 (**A-d**) or IL17 (**B-e**). Cells were quantitatively evaluated by Image-Pro Plus. The data are shown as the mean ± SD of three replicates, where * represents *p* < 0.05, ** represents *p* < 0.01, and *** represents *p* < 0.001.

**Figure 10 microorganisms-08-00195-f010:**
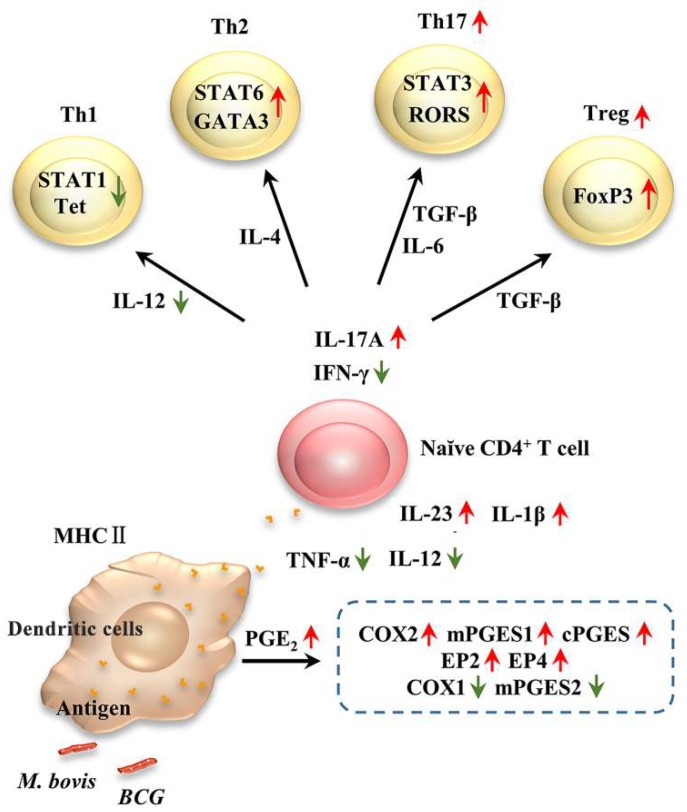
Differentiation of CD4^+^ T cell subsets under the corresponding cytokine atmosphere stimulated by *M. bovis* and BCG infected DCs. Arrows represent the main significantly differential expression of molecules in DCs and CD4^+^ T cell subsets induced by infection with *M. bovis* compared with BCG: green arrow indicating down-regulation, while red arrows up-regulation.

**Table 1 microorganisms-08-00195-t001:** The primer sequences of qRT-PCR.

Name		Primers (5’→3’)		
*β-actin*	F	GGCTGTATTCCCCTCCATCG	R	CCAGTTGGTAACAATGCCATGT
*COX-1*	F	CCGAGGAGCCAGCCGTTG	R	AGCCCTGTATTCCGTCTCCTT
*COX-2*	F	GTGCTGGAAAAGGTTCTTCTACG	R	GTGAACCCAGGTCCTCGCTT
*mPGES1*	F	CTGCTGGTCATCAAGATGTACG	R	CCCAGGTAGGCCACGGTGTGT
*mPGES2*	F	CCTACAGGAAAGTGCCCATCT	R	CCACTTCATCTCCTCCGTCC
*cPGES*	F	TGGGAGGATGACTCAGATGAAG	R	TCCAGGCGATGACAACAGC
*ROR-γt*	F	CAGTCTACATGCAGAAGTGC	R	ATGTAAGTGTGTCTGCTCCG
*Foxp3*	F	CCCATCCAATAAACTGTGGTCA	R	CTCTCTTTCATTTGGTATCCGCT
*T-bet*	F	CCTGGACCCAACTGTCAACT	R	AACTGTGTTCCCGAGGTGTC
*GATA3*	F	GAAGGCATCCAGACCCGAAAC	R	ACCCATGGCGGTGACCATGC
*IFN-γ*	F	GCTCTGAGACAATGAACGCTAC	R	GCTCTGAGACAATGAACGCTAC
*IL-1β*	F	CAACCAACAAGTGATATTCTCCATG	R	GATCCACACTCTCCAGCTGCA
*IL-6*	F	TGATGCACTTGCAGAAAACA	R	ACCAGAGGAAATTTTCAATAGGC
*IL-17*	F	TCCAGAAGGCCCTCAGACTA	R	AGCATCTTCTCGACCCTGAA
*TNF-α*	F	CTCCATCAACAGCCCTCTGG	R	GAGGGCATTGGCATACGAGT

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
