# Peer review of "Upregulation of Cytokines and Differentiation of Th17 and Treg by Dendritic Cells: Central Role of Prostaglandin E2 Induced by *Mycobacterium bovis"

_microorganisms, 2020, doi:10.3390/microorganisms8020195_

Round 1
Reviewer 1 Report
Interesting and original paper , innovative job and absolutely well done.
TB it' s really important disease included most important role in one health management and direction and approach done in this paper it was correct in lab direction guaranteed in statiistical analysis too and in results guaranteed in this paper.
Introduction ,materials and methods , included DC preparation and bacterial culture, detecion and use of PGE2 with ELISA procedure, genes transcription concerning PGE2, western blot Cox assay and mouse infection and histopathology have been well done and correctly presented.
Conduction line and results are balanced and scientifical results clear in ESB objective.
Author Response
Special thanks to you for your good comments.
The manuscript has undergone English language editing by MDPI. The text has been checked for correct use of grammar and common technical terms, and edited to a level suitable for reporting research in a scholarly journal.
Please see the attachment.

Reviewer 2 Report
Liu et all., discuss the differential effect of M. bovis and BCG on the induction T regs and production of different cytokinesand the role of a cyclooxygenase dependent mechanism. The authors used in vivo and in vitro experiments to demonstrate the differential role of the pathogens and their interaction with APCs. There are some key concerns and they are listed below.
The authors claims the in vitro generated cell subsets are dendritic cells. The bone marrow (GMCSF or GMCSF+ IL4) culture system can generate a mixture of macrophages and dendritic cells (PMID: 26084029). So the authors has to demonstrate some specific markers to demonstrate that they are not a mixture of cells and also the relative frequency of each subsets (please use additional markers as detailed PMID: 27637149).
The authors mention that the histopathology of spleen shows increased macrophages and that also correlate with the intensity of the lesions. Hence the macrophages in the mixed cultures may be a key component for the observed phenomenon.
So authors has to clearly demonstrate that which cell subsets is infected with mycobacterium, rather than depending on a reports published in ( Inaba et al. (1992) and Lutz et al. (1999) ).
Otherwise the authors should use a general term mononuclear phagocytes instead of dendritic cells in the article. In a physiological condition the subsets may not be important for the infection by the pathogens but it is important for readers to understand the role each APC subsets and involvement in regulating the infections.
Figure 2 and Fig 3 the relative expression of COX2 and PGES are decreased with Indometacin treatment and does the observation is reflected in the protein level?
The treatment reduce different inflammatory cytokines such as IL17, IL23, TNFa etc, it will be great to see the expression of IL1β level at protein level. The IL1β requires specific processing and maturation steps and hence the change at a transcriptional level may not influence the biological outcome. The authors also mention that IL-6 is not changed at RNA level, if they are changed at protein level, again indicate a cyclooxygenase independent mechanism. So the observed phenomenon may be an inhibition of inflammatory pathways by Indometacin than a COX dependent pathway.
To prove that if the authors use any TLR agonists like LPS, CPGs, any other TLR2, TLR5 or TLR7, or synthetic TLR agonist and treat the mononuclear phagocytes in the presence and absence of Indometacin. If you observe the same phenomenon, it clearly demonstrate the independent role of Indometacin.
The authors demonstrate the differential kinetics of bacterial load and T Reg development in the infected animals. It will be critical to purify the different MNP subsets, especially the two cDC subsets, pDC and macrophages and check for the cyclooxygenase and other cytokine mRNAs to confirm the in vitro model reflect what authors observe with the in vitro system.
Does the authors administrated Indometacin and observe similar to that of in vitro results?
Author Response
Those comments are all valuable and very helpful for revising and improving our paper, as well as the important guiding significance to our researches.
We have studied comments carefully and have made correction which we hope meet with approval.
Please see the attachment.

Round 2
Reviewer 2 Report
Dear Authors
The additional information make the article better and there is a minor suggestion on Figure 5, the MHC II- PE vs FITC and what is the antigen on FITC ? Please add the details.
thanks